# Heterozygous *NPR2* Variants in Idiopathic Short Stature

**DOI:** 10.3390/genes13061065

**Published:** 2022-06-15

**Authors:** Lana Stavber, Maria Joao Gaia, Tinka Hovnik, Barbara Jenko Bizjan, Maruša Debeljak, Jernej Kovač, Jasna Šuput Omladič, Tadej Battelino, Primož Kotnik, Klemen Dovč

**Affiliations:** 1Clinical Institute for Special Laboratory Diagnostics, University Children’s Hospital, UMC, 1000 Ljubljana, Slovenia; lana.stavber@kclj.si (L.S.); tinka.hovnik@kclj.si (T.H.); barbara.jenko.bizjan@kclj.si (B.J.B.); marusa.debeljak@kclj.si (M.D.); jernej.kovac@kclj.si (J.K.); 2Department of Pediatric Endocrinology, Diabetes and Metabolic Diseases, University Children’s Hospital, UMC, 1000 Ljubljana, Slovenia; jasna.suputomladic@kclj.si (J.Š.O.); tadej.battelino@mf.uni-lj.si (T.B.); primoz.kotnik@mf.uni-lj.si (P.K.); 3Hospital Center Vila Nova de Gaia Espinho, 4430 Porto, Portugal; maria.stgaia@gmail.com; 4Institute of Biochemistry and Molecular Genetics, Faculty of Medicine, University of Ljubljana, 1000 Ljubljana, Slovenia; 5Department of Paediatrics, Faculty of Medicine, University of Ljubljana, 1000 Ljubljana, Slovenia

**Keywords:** short stature, *NPR2* gene, small for gestational age, growth hormone therapy

## Abstract

Heterozygous variants in the *NPR2* gene, which encodes the B-type natriuretic peptide receptor (NPR-B), a regulator of skeletal growth, were reported in 2–6% cases of idiopathic short stature (ISS). Using next-generation sequencing (NGS), we aimed to assess the frequency of *NPR2* variants in our study cohort consisting of 150 children and adolescents with ISS, describe the *NPR2* phenotypic spectrum with a growth pattern including birth data, and study the response to growth hormone (GH) treatment. A total of ten heterozygous pathogenic/likely pathogenic *NPR2* variants and two heterozygous *NPR2* variants of uncertain significance were detected in twelve participants (frequency of causal variants: 10/150, 6.7%). During follow-up, the *NPR2* individuals presented with a growth pattern varying from low–normal to significant short stature. A clinically relevant increase in BMI (a mean gain in the BMI SDS of +1.41), a characteristic previously not reported in *NPR2* individuals, was observed. In total, 8.8% participants born small for their gestational age (SGA) carried the *NPR2* causal variant. The response to GH treatment was variable (SDS height gain ranging from −0.01 to +0.74). According to the results, *NPR2* variants present a frequent cause of ISS and familial short stature. Phenotyping variability in growth patterns and variable responses to GH treatment should be considered.

## 1. Introduction

Idiopathic short stature (ISS) is defined as a height that is more than two standard deviations (SD) below the average height for race, age and gender, after excluding other systemic, endocrinal, nutritional or chromosomal abnormalities [1]. ISS is the most common form of short stature, accounting for 60–80% of cases, but its specific etiology is, in most cases, unclear [2].

Final height is highly heritable, with more than 500 loci associated with human height have been identified by genome-wide association studies [3]. With the development of genetic testing technology, an increasing number of novel monogenic causes of ISS have been recently proposed [3,4,5]. It is possible to identify a causative gene in 25–40% of children with ISS, highlighting that this is a heterogeneous condition with different etiologies, many of which are genetic [3,5].

One of the genetic causes that has been reported includes variants in the *NPR2* gene, located on chromosome 9p13.3 and encoding the B-type natriuretic peptide receptor (NPR-B), which plays an important role in the complex paracrine regulation of the growth plate [2,3,6]. The *NPR2* signaling cascade promotes bone matrix synthesis and stimulates the proliferation and differentiation of cartilage; thus, alterations in that pathway lead to growth disorders [2,7].

Biallelic pathogenic variants in the *NPR2* gene cause severe skeletal dysplasia, acromesomelic dysplasia type Maroteaux, which is generally characterized by a height SDS of <−5, bone deformities and other stigmata [4,6,7,8,9]. On the other hand, heterozygous pathogenic *NPR2* variants lead to milder growth disorders, mostly classified as ISS, although some of the affected individuals might present mild signs of bone dysplasia [2,4,5,10].

Through systematic phenotyping of patients with short stature, our cohort study included 150 participants with ISS. Using next-generation sequencing (NGS), we aimed to assess the prevalence of *NPR2* variants among children with ISS, characterize the phenotypic spectrum with a growth pattern including birth data, and study the response to growth hormone treatment.

## 2. Materials and Methods

### 2.1. Participants

One hundred and fifty children and adolescents (64 females, aged between 3 and 20) with short stature (i.e., height below −2 standard deviation scores (SDS)), referred to the University Children’s Hospital of Ljubljana between 2015 and 2021, were enrolled in this study. We excluded children/adolescents with the following: (i) growth hormone deficiency, (ii) hypothyroidism, (iii) chronic illness, (iv) defined skeletal dysplasia and/or syndrome, (v) cytogenetically detectable chromosomal abnormalities (e.g., Turner syndrome) and (vi) growth-influencing medications (glucocorticoids) (Appendix A).

The study was conducted according to the guidelines of the Declaration of Helsinki and approved by the Slovene Medical Ethics Committee (0120-36/2019/4). All participants or their legal guardians provided informed consent before enrollment.

### 2.2. Methods

#### 2.2.1. Clinical Procedures

Clinical data of the probands were obtained from their medical history and medical documentation via electronic medical records. Arginine and L-Dopa GH stimulation tests were performed according to previously published test procedures [11]. Serum GH levels were determined by immunoassay using Immulite 2000 (Siemens Healthcare Diagnostics, Tarrytown, NY, USA). Bone age was evaluated based on the Greulich and Pyle (GP) atlas bone age determination system, 2nd edition, or determined with the BoneXpert program [12]. Z-scores for height were calculated using the LMS method and the British 1990 growth reference data [13].

#### 2.2.2. Laboratory Procedures

##### DNA Isolation

Whole-blood EDTA samples from all study participants and additionally a buccal swab from participant no. 5 (P5) (because the blood sample indicated a mosaic form of the *NPR2* variant) were collected for isolation of genomic DNA according to established laboratory protocols with the FlexiGene DNA isolation kit (Qiagen) [14].

##### Next-Generation Sequencing and Variant Interpretation

In all participants, clinical exome (TruSight One) (n = 67) or whole exome sequencing (WES) (n = 83) was performed according to availability. NGS library preparation was undertaken according to standard Illumina protocols (Illumina DNA Prep with Enrichment). The NGS libraries for clinical exome sequencing were prepared using the Illumina TruSight One sequencing panel and sequenced on the MiSeq Illumina Platform sequencer (both Illumina, San Diego, CA, USA). WES sequences were generated on an Illumina NovaSeq 6000 System. For bioinformatic analyses, a bcbio-nextgen workflow toolkit was used (https://bcbio-nextgen.readthedocs.io/ accessed on 14 June 2022). Reads were aligned to the GRCh37 assembly of the human genome with the BWA-mem [15], using samtools [16] and sambamba [17] to sort bam files and mark duplicate reads. Variant calling was performed according to GATK Best Practices Workflows for small germline variants calling with HaplotypeCaller [18]. VarAFT software was used for variant annotation and filtering [19]. Copy number variations in the ROI (region of interest) were inferred by the CNVkit Python library [20].

The minor allele frequency threshold for known variants was set at 1%, and all variants exceeding this value were excluded from further analysis. All variants were classified following the American College of Medical Genetics and Genomics/Association for Molecular Pathology (ACMG/AMP) variant pathogenicity guidelines [21].

##### Sanger Sequencing

A confirmation variant analysis with Sanger sequencing was only needed in P5 to study the mosaic form of the determined variant. Targeted Sanger sequencing was also needed for segregation analysis.

The whole study protocol is schematically shown in Appendix A.

## 3. Results

Among 150 study participants, the NGS analysis revealed a total of 10 causal heterozygous *NPR2* variants in 10 probands (10/150, 6.7%) and 2 heterozygous *NPR2* variants of unknown significance (VUS) in 2 additional probands. Six *NPR2* variants were novel (in P1, P2, P4, P5, P10, and P11) and six *NPR2* variants were previously reported in the existing genomic variant databases (ClinVar, HGMD) (in P3, P6, P7, P8, P9, and P12) (Table 1). According to the ACMG/AMP guidelines [21], the nonsense variants (c.595C>T, c.1571dupA, c.2761C>T, and c.844C>T) were classified as pathogenic (ACMG criteria: PVS1, PM2, PP3, and PP1), whereas the missense variants as likely pathogenic (c.1636A>T, c.2644G>A, and c.2633C>T; ACMG criteria: PP1, PM2, PP3, and BP1) or as a variant of unknown significance (c.532C>T and c.1517G>A; ACMG criteria: PM2, PP3, and BP1). NGS and consequent Sanger sequencing from the blood and buccal sample indicated that the variant c.532C>T was mosaic, with approximately 30% of altered nucleotide in both samples. The variant c.2644-1G>A affects a donor splice site in intron 3 of the *NPR2* gene. It is expected to disrupt RNA splicing and likely results in an absent or disrupted protein product. Donor and acceptor splice site variants typically lead to a loss of protein function [22], and loss-of-function variants in *NPR2* are known to be pathogenic [23,24]. The currently available evidence and ACMG criteria indicate that the variant is pathogenic, but additional data are needed to verify that conclusively. Therefore, we decided to classify this variant as likely pathogenic.

All reported variants, except in P5 and P8 where segregation analysis was not feasible, were inherited. In all *NPR2* families, the reported variant co-segregated with the *NPR2* phenotype (range of parents’ height SDS: −2.76 to −0.6), except in P12′s family, where the variant was inherited from the mother, who had normal height (the reported variant is, therefore, classified as VUS) (Table 1).

### Phenotypic Characteristics of Individuals with Heterozygous NPR2 Variants

In total, 12 probands (9 males, 3 females) with a heterozygous *NPR2* variant from 12 unrelated families were determined (Table 1). Their age ranged from 3 to 17. The average birth weight of the probands (*n* = 11) was −1.23 SDS (range: +0.48 to −2.32) and the average birth length (*n* = 11) was −1.32 SDS (range: +0.02 to −2.26) (the SDS data for P12, who was born within 30 weeks of gestation, was not calculated because of LMS limitations, but the birth data were normal). Out of the 12 probands, 3 were born small for their gestational age (SGA) (i.e., the birth length or/and weight ≤ −2 SDS [25]; P3, P4, and P11), whereas 4 were born with a low–normal birth weight or length (i.e., −1.5 SDS to −2 SDS; P5, P6, P7, and P10) (Table 1). The height SDS of individuals with the *NPR2* variant (including parental data) ranged from low–normal to significant short stature (range: −0.6 to −3.61 SDS). The mean height and BMI SDS values for all *NPR2* individuals calculated for age, presenting an average height and BMI trend, are illustrated in Figure 1. The weight and BMI SDSs of the 12 probands ranged from significantly low to normal (range: −3.98 to +0.72 SDS for weight and −3.84 to +1.98 SDS for BMI). The BMIs had an increasing pattern during follow-up, with a mean gain of +1.41 SDS from birth to the last evaluation (range: −0.74 to +2.96 SDS). At birth, 10/12 probands had a BMI SDS between 0 and −2.35 and, at the last evaluation, 9/12 were above 0 SDS. In total, 9/12 probands had delayed bone age (range: −1.88 to −0.15 SDS), whereas in 3/12 (P2, P6, and P12), the probands bone age was advanced (range: +0.42 SDS to +1.44 SDS).

In three probands (P4, P5, and P7), we found additional skeletal findings, such as a shortened metacarpal bone of the fourth and fifth finger, a slightly shortened proximal part of the lower and upper extremities, clinodactyly and micrognathia (Table 1).

Three probands (P3, P7, and P10) met national guidelines for growth hormone treatment and were treated with recombinant human growth hormone (rhGH) (dose (range): 35–50 μg/kg/day; duration (range): 8.5–68 months). P3 and P7 also received therapy with GnRH analogue (triptorelin embonate). Significant height gain was observed in P7 and P10. P7 started rhGH treatment at 10.75 years of age and had a height gain of +0.74 SDS in 5 years and 8 months (from −3.14 to −2.4 SDS). P10 started rhGH treatment at 5.9 years of age and had a height gain of +0.49 SDS in 8 months (from −2.56 to −2.07 SDS). In P3, therapy with rhGH and GnRH analogue was introduced at 13.25 years of age. After 13 months of therapy, there was no change in growth velocity (height SDS −2.16 to −2.17). The peak GH values, height gain, age at introduction of GH therapy and duration of therapy are shown in Table 2.

## 4. Discussion

The present study, including 150 participants with ISS, demonstrated a 6.7% (10/150) total yield of causal *NPR2* variants (likely pathogenic or pathogenic). We identified *NPR2* causal variants in 11.6% of familial cases (regarding our whole study cohort of 150 participants). Recently, heterozygous *NPR2* variants were reported in 2–6% of ISS cases [4,10] and in 13.6% of familial ISS cases [26]. Additionally, besides the (likely) pathogenic *NPR2* variants, our study revealed two patients (P5 and P12) with a currently classified variant of uncertain significance, and in P5, in a likely mosaic form. All reported variants were inherited (except in P5, where it is assumed to be de novo, and in P8, where segregation analysis was not feasible). All reported *NPR2* variants co-segregated with the *NPR2* phenotype, except in the family of P12, where the variant was inherited by a mother of normal height (−0.11 SDS); therefore, the variant was classified as (cold) VUS (Table 1).

In our study, the severity of short stature and the presence of nonspecific skeletal abnormalities varied across the *NPR2* individuals with a range of height SDSs between low–normal (−0.6 SDS) and severe short stature (−3.61 SDS), including the data of the final height SDSs of the *NPR2* parents (Table 1). Our results also indicated that in carriers of heterozygous causal *NPR2* variants, short stature can be progressive (Figure 1), consistent with previous observations [7]; however, progressive growth deficit was not observed in all our *NPR2* individuals. We did not find a specific correlation between the type of variant (nonsense vs. missense) and the growth pattern or final height of the *NPR2* carriers. The severity of short stature is likely attributed to variable expressivity and additional unknown multifactorial genetic and environmental factors, even within the same family.

Importantly, our study showed a progressive and clinically relevant increase in the BMIs during follow-up (Figure 1), with a mean gain in BMI SDS of +1.41. Previous reports showed that *NPR2* heterozygous variant carriers have, in general, a normal BMI [4,7,10,27]. No study has previously shown this specific pattern for *NPR2* individuals. It has been reported that the C-type natriuretic peptide (CNP), for which the NPR-B is a receptor (encoded by the *NPR2* gene), is a natural regulator of adipogenesis, playing a role in the development of obesity in childhood [28]. It acts as a suppressor of obesity, and its level was decreased in the obese groups compared to controls, suggesting a defective natriuretic peptide system in these individuals [28,29]. It was also shown that CNP suppresses obesity in mice [30]. This might help to explain the BMI’s tendency to increase as shown in this subset of patients with impaired NPR-B function; however, to clarify this hypothesis, the percentage of body fat should be measured in the *NPR2* individuals, which is missing in the current study. Additionally, an increase in BMI could also be due to maintenance or a decrease in height SDSs, while weight SDSs mainly increase.

Twenty-five percent of *NPR2* heterozygous carriers (3/12) in our cohort were born SGA, which accounts for 8.8% of all SGA in our whole study cohort (3/34). Due to the heterogeneous pathogenesis underlying fetal and postnatal growth restriction in SGA children, current evidence suggests that a considerable number of SGA patients without catch-up growth carry monogenic, polygenic, or epigenetic variants, which explain their growth failure [25]. Several case reports and candidate gene studies have demonstrated that skeletal dysplasia accounts for more than 20% of short stature in both SGA and ISS children [31], whereas other known monogenic causes explaining SGA and ISS, such as *IGF1*, *IGF1R* and *SHOX* variants, present a smaller percentage of cases [32]. According to the limited published information, *NPR2* variants could present the cause of SGA without catch-up growth [6,25,27,33], but such a connection currently seems to be mainly based on observed *NPR2* phenotyping characteristics. Freire et al. found two *NPR2* variants among 55 non-syndromic SGA patients, which represented 3.6% of the studied SGA population [27], whereas Plachy et al. reported a 6% yield of *NPR2* variants in children born SGA among the cohort of familial short stature [6].

Even though SGA is one of the indications for rhGH therapy, it remains unclear whether carriers of *NPR2* variants, being SGA or not, respond to rhGH therapy. Data were available on growth hormone response in three of our participants with pathogenic/likely pathogenic *NPR2* variants (P3, P7, and P10). The response to GH treatment was variable, with a height SDS gain ranging from −0.01 to 0.74 (Table 2). The two children in whom rhGH therapy was introduced earlier (5.9 and 10.75 years) showed a relatively good response, with a mean gain of 0.62 SDS (0.49–0.74), even accounting for the short duration of treatment for proband P10 (8 months). There was no change in growth velocity in P3 after 13 months of treatment with rhGH. However, it must be noted that the therapy was started at 13.25 years of age, the individual was simultaneously also receiving GnRH analogue, and the final height is yet to be determined. Even though the response to treatment seems promising in two out of three treated participants, it is still difficult to assess the efficacy, due to various courses of treatment (different ages of therapy initiation, different duration, absent data of final height, etc.). Nevertheless, growth cessation was not observed in any of the treated patients, which contrasts with our non-treated *NPR2* carriers. The effect of GH treatment in individuals with *NPR2* variants has been described in small case series with different inclusion criteria and showed conflicting results [2,4,5,6,10]. Vasques et al. reported a relatively poor response to rhGH treatment (height gain of 0 to 0.4 SDS), but the initiation of treatment was after puberty onset, and despite a mean duration of treatment of 3.6 years, final height data were only available for one individual [10]. Other studies reported quite a promising response to rhGH, being similar to the outcomes in individuals with SHOX deficiency, with height improvements of +1.2 to 2.46 SDSs with long-term treatment of 2 to 9 years; however, data on final heights are mostly missing [2,4,6,34]. Due to different courses of treatment in the previous and current studies, various durations of follow-up during treatment, various ages of initiation, additional therapies, and lack of data on final heights, currently, it is still difficult to evaluate GH efficacy in *NPR2* individuals.

## 5. Conclusions

In conclusion, our findings confirm that *NPR2* variants present a frequent cause of ISS and familial short stature, but the growth pattern can vary from low–normal to significant short stature, even within the same family. Our study showed clinically relevant increases in BMIs in *NPR2* carriers, although further investigation, including measures of body fat, should be conducted to clarify the role of *NPR2* variants in obesity risk. Among all the SGA children in our study cohort, 8.8% carried the *NPR2* causal variant, highlighting possible prenatal causes of intrauterine growth restriction, especially within familial cases. As the current study presented varied responses to growth hormone treatment with limited data concerning efficacy, additional larger studies with earlier initiation and long-term treatment with data on final heights are needed. Finally, our results expanded the number of classified *NPR2* variants, enabling easier variant interpretation for future cases.

## Figures and Tables

**Figure 1 genes-13-01065-f001:**
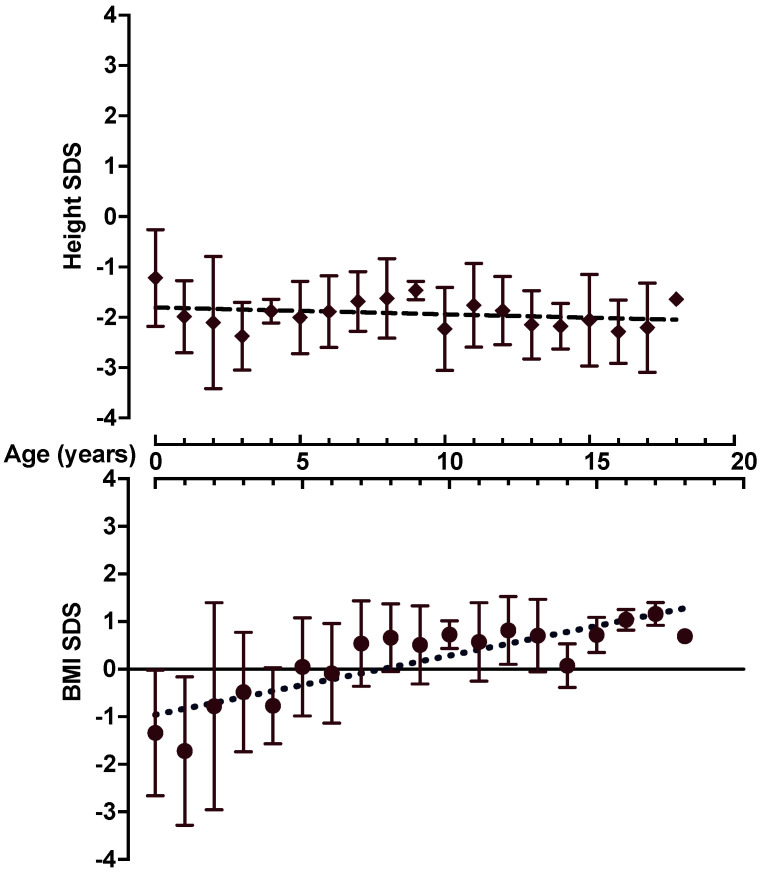
Mean height and BMI SDS trend related to age for all *NPR2* probands.

**Table 1 genes-13-01065-t001:** Clinical features of study probands at current age and final height of their parent with the corresponding heterozygous *NPR2* variant. f—female, m—male, yrs—years, N/A—not available, *NPR2* parent—parent carrying *NPR2* variant * GRCh37, NM_003995.4, NP_003986.2, NG_009249.2.

Participant (No.)	Gender (f/m)	Age (yrs)	Height (cm)	Height (SDS)	Weight (SDS)	BMI (SDS)	Birth Weight (SDS)	Birth Length (SDS)	GH Therapy (Yes/No)	Additional Skeletal Findings	Final Height of NPR2 Parent	Heterozygous Mutation in *NPR2* Gene *
P1	m	12.13	140.6	−1.17	0.11	1.0	0.48	−0.48	no	/	−2.76	c.595C>T, p.Gln199Ter
P2	m	12.93	132.1	−2.84	−0.45	1.6	0.22	−0.1	no	/	−2.06	c.1571dupA, p.Tyr524Ter
P3	m	14.35	146.8	−2.17	−1.72	−0.63	−2.14	−1.61	yes	/	−1.91	c.1636A>T, p.Asn546Tyr
P4	f	14.15	150.2	−1.53	−0.47	0.5	−2.32	−2.26	no	Shortened proximal part of lower and upper extremities, shortened metacarpal bone of 4th finger	−0.6	c. 2644-1G>A
P5	m	8.59	124	−1.26	−1.06	−0.41	−1.58	−1.89	no	Micrognathia, clinodactyly	N/A	mosaic c.532C>T, p.Arg178Trp
P6	f	17.35	146	−2.83	−0.75	0.99	−1.87	−1.96	no	/	−2.2	c. 2644G>A, p.Val882Ile
P7	m	16.5	156.9	−2.43	−0.79	0.87	−1.22	−1.52	yes	Shortened metacarpal bone of 4th and 5th finger, clinodactyly	−0.6	c. 2761C>T, p.Arg921Ter
P8	m	14.14	144	−2.33	−1.35	0.11	−0.85	−1.08	no	/	N/A	c. 844C>T, p.Gln282Ter
P9	m	17.52	165.1	−1.64	−0.45	0.69	−1.05	−0.02	no	/	−1.43	c. 2761C>T, p.Arg921Ter
P10	f	6.71	109.4	−2.0	−2.07	−1.07	−1.46	−1.61	yes	/	−1.6	c.595C>T, p.Gln199Ter
P11	m	3.54	92.6	−1.74	−2.24	−1.44	−1.7	−2.02	no	/	−1.76	c.2633C>T, p.Thr878Ile
P12	m	13.17	149	−0.88	0.19	0.94	N/A	N/A	no	/	−0.11	c.1517G>A, p.Arg506His

**Table 2 genes-13-01065-t002:** GH stimulation testing with arginine and/or L-dopa results and growth follow-up in all participants receiving GH.

Proband	Peak GH Arginine (μg/L)	Peak GH L-DOPA (μg/L)	Height before GH (SDS)	Height with GH (SDS)	Age at GH Introduction (Years)	GH Therapy Duration (Months)
P3	/	/	−2.16	−2.17	13.27 *	13
P7	16.2	/	−3.14	−2.43	10.78 **	68
P10	/	9.38	−2.64	−2.00	6	8.5

* Simultaneous introduction of GnRH analogue therapy. ** From age 12.2 to 14.56, P7 received additional GnRH analogue therapy (i.e., 28 months).

## Data Availability

Available from corresponding author on reasonable request.

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
