# Peer review of "Heterozygous NPR2 Variants in Idiopathic Short Stature"

_genes, 2022, doi:10.3390/genes13061065_

Round 1

Reviewer 1 Report

Paper for review

Stavber et al. report on Heterozygous NPR2 variants in idiopathic short stature – a 2 comprehensive clinical aproach to NPR2 probands.

The clinical and molecular data very are very interesting However, there some minor concerns that need to be addressed.

General comments:

  1. It would be better to divide the Methods into small paragraphs with headings.
  2. Flow sheet diagram would be helpful for the readers to understand the overall methodology.
  3. The authors use the term "mutation(s)". HGVS recommends the use of the neutral term "variant(s)"
  4. The reference sequence used should be described properly, the version number (e.g. NG_0123456. 3; NP_5462.2) is missing. ** Please note that since the reference sequence given (NM_) does not contain intronic sequences a genomic reference sequence should be given as well (e.g. NG_0123456.3).
  5. Gene name italic (title).
  6. As it’s a well-known disorder, do the authors have any patient’s images? So that we would have an idea about the disease phenotype.
  7. Any phenotype-genotype correlation associated with the position of the mutation in the NRP2
  8. Gain of function in this gene causes long stature, any case observed in your cohort?
  9. Kindly cite: PMID: 30359775, PMID: 30016695
  10. What about VUS, they might cause some phenotypic variations in patients, please comment.

Reviewer 2 Report

The paper  is clear, the methods appropriate, the discussion good.

Author Response

Thank you for Your positive comments.

Reviewer 3 Report

In this paper, the authors assess the frequency of NPR2 variants and phenotypic spectrum with growth pattern in idiopathic short stature cohort consisting of 150 children and adolescents. 

The following issue question should be addressed:

Authors described NPR2 variants were co-segregated in family members and associated with final height of parents in several places. However, the clinical phenotypes and NPR2 genotypes of parents or family members were missing. 

Line 124, All reported variants, except in P5 in P8 where segregation analysis was not feasible, were inherited. In all positive families the reported variant co-segregated with NPR2 phenotype (range of parents’ height SDS:-2.76 to -0.6), except in P12 family, where the variant is inherited by mother, who had normal height. 

Line 181, All reported NPR2 variants co-segregated with NPR2 phenotype, except in family of P12, where the variant was inherited by mother with normal height (-0.11 SDS) and the variant was therefore classified as (cold) VUS.

Line 185, including the data of final height SDS of NPR2 positive parents.

Line 250, NPR2 variants present a frequent cause of 249 ISS and familial short stature, but the growth pattern can vary from low-normal to signif-250 icant short stature, even within the same family.

Line 197-203, BMI dose not measure body fat directly. As authors would like to make a major conclusion of that NPR2 carriers could affect regulation of adipogenesis in these individuals (Lin, 252), body fat percentage should be measured in the NPR2 families or probands to clarify the roles of NPR2 variants in obesity risk.  

In title: Heterozygous NPR2 variants in idiopathic short stature – a comprehensive clinical aproach to NPR2 probands

This reviewer is confused about the meaning of “a comprehensive clinical approach to NPR2 probands”.

No clinical approach is proven for prediction or treatment of NPR2 probands in this study

There are some minor issues: 

Line 17, ISS spell out in the Abstract. 

Line 137, NPR2 positive? You mean variants? 

Round 2

Reviewer 3 Report

The authors have adequately addressed my previous comments.